# Physical as Well as Psychological Distress and Coping with Situational Dilemmas Experienced by People Infected with COVID-19: A Mixed Method Study

**DOI:** 10.3390/ijerph192214657

**Published:** 2022-11-08

**Authors:** Arunee Hengyotmark, Wichitra Kusoom

**Affiliations:** 1Kuakarun Faculty of Nursing, Navamindradhiraj University, Bangkok 10300, Thailand; 2Faculty of Nursing Science, Bangkokthonburi University, Bangkok 10170, Thailand

**Keywords:** COVID-19, physical distress, mental distress, coping, mixed methods study, Thailand

## Abstract

The COVID-19 pandemic caused serious health problems that affected people around the globe. This study aims to understand the physical distress (PhyD), psychological distress (PsyD), and coping experiences among people infected with COVID-19, develop a grounded theory, and examine PhyD, PsyD, and coping among people infected with COVID-19. A sequential exploratory mixed methods strategy is employed. A qualitative procedure is based on a grounded theory; data collection includes observation and in-depth interviews with 25 participants, aged 18 years and above. The quantitative one included 180 participants. Content analysis was applied using the Strauss and Corbin method, and ATLAS.ti software. Descriptive statistics, Pearson’s correlation, and the independent t-test were used. Results: The six major themes, including (1) severity of COVID-19 symptoms, (2) death anxiety, (3) uncertainty, (4) barrier to healthcare access, (5) compliance and self-regulation coping (6) post-COVID-19 effects. PhyD, PsyD, and coping were all at a moderate level. The relationship between PhyD, PsyD, and coping was positive. The prevalence in post-COVID-19 effects was 70% (95% CI 63.3-76.4%). There were higher amounts in women than men. The most frequent residual symptoms were decreased activity tolerance (40%), fatigue (33.3%), anxiety and fear of abnormal lungs (33.3%), dyspnea (27.8%), allergy (24.4%), and lung impairment (22.2%). Moreover, the prevalence of more than two symptoms was 54% (95% CI 47.2–61.7%). This study considers that the healthcare providers should be concerned with sufficient healthcare services. Interventions are needed for supporting their recovery from COVID-19 effects.

## 1. Introduction

The novel coronavirus (COVID-19) pandemic caused serious health issues that affected people physically and psychologically and which is still spreading around the globe. Similarly, Thailand’s population of 66.1 million [1] has approximately 4.4 million confirmed COVID-19 cases with 29,844 deaths [2]. The Ministry of Public Health, Thailand, reported that the situation of COVID-19 in Thailand during January–30 May 2022, reached 2.2 million cases. Metropolitan Bangkok is the capital city that had the highest number of COVID-19 cases from January–30 May with 355,956 cases. Moreover, the northeastern area included 46,047 cases in Nakhon Ratchasima province, 37,892 cases in Ubon Ratchathani province, and 13,150 cases in Nakhon Phanom province [3]. The novel COVID-19 is caused by severe acute respiratory syndrome coronavirus 2 (SARS-CoV-2) [4]. It can spread through droplets and human-to-human transmission and by indirect contact or contaminated objects and airborne particles [5]. As a result of the pandemic, patients have had experiences with physical and mental distress. The symptoms include fever, chills, headache, dyspnea, cough, loss of taste, sore throat, body ache, etc. A systematic review and meta-analysis reported that fever was the most common symptom at 88.47%, which was followed by cough, fatigue, and, less proportionally, dyspnea and myalgia. Dyspnea was the only symptom, which was associated with the severity of COVID-19 [6]. These were more than 50% of the patients and developed within four days in the above 90%, and shortness of breath was significantly higher in males [7]. The study revealed that the COVID-19 pandemic affected more than two-thirds (69%) who experienced moderate-to-very high levels of psychological distress, which was 46% in Thailand and 91% in Egypt [8]. In Thailand, the number of male patients was higher than female [9]. The 25.7% of patients with COVID-19 still suffered from psychological distress, which should receive timely attention from healthcare workers, and the severity of the disease and disease uncertainty has a significant impact on distress [10]. A systematic review and meta-analysis reported that the global prevalence estimate was 28% for depression, 26.9% for anxiety, 24.1% for post-traumatic stress symptoms, 36.5% for stress, 50% for psychological distress, and 27.6% for sleep problems [11]. Acute respiratory distress syndrome (ARDS) is a serious complication of COVID-19. The study found that overall prevalence of ARDS was 3.6%: 2.9% among females and 4.4% among males [12]. Many patients are critically ill, and ventilators are needed. The Public Health Ministry of Thailand is encouraging patients with COVID-19 to recuperate at home in response to the kingdom’s bed shortage [13]. The people with the COVID-19 infection have to stay for some time in an isolation setting in hospital. Because of this, their emotional reaction was negative as they were perceived as a threat by others [14]. While the measures taken by the government slowed down the spread of the pandemic in Thailand, they resulted in loss of jobs, incomes, businesses, and food security for families and education for children [15]. In Thailand, the COVID-19 pandemic impacted nurses in four ways. These were: (1) work–life imbalance because of increased workload, (2) fear of infection and transmission, (3) inadequate organization support including supply of personal protective equipment (PPE) and quality vaccines, information support, and unfair compensation in some hospitals, and (4) ecological changes in both positive and negative directions [16]. The prevalence of the COVID-19 pandemic is a situational dilemma crisis in Thailand. Unfortunately, there were no mixed method studies to understand physical as well as psychological distress, and coping experiences among people infected with COVID-19. A mixed methods study using a sequential exploratory design was employed. A qualitative procedure based on a grounded theory. It used quantitative data to assist in the interpretation of qualitative findings. Finally, the development of an instrument is needed because an existing instrument is not available [17]. A grounded theory is a strategy of inquiry in which the researchers derived a general, abstract theory of process and action grounded in the view of participants and is highly useful for uncovering [17,18,19]. It is a real experience of people with the COVID-19 infection. 

Therefore, this study aimed to (1) understand the physical, psychological distress, and coping experiences, (2) develop a grounded theory, and (3) examine physical, psychological distress, and coping experiences. 

## 2. Materials and Methods

### 2.1. Study Design and Participant Recruitment

A mixed methods study using a sequential exploratory design was employed; the qualitative part was emphasized more than the quantitative one. It included a three-phase approach.

The finding from the initial qualitative phase was to build an instrument and focuses on the subsequent quantitative phase [17]. A qualitative procedure was based on a grounded theory because it is to move beyond description and discover a theory [17]. For a quantitative procedure, the sequential exploratory supportive data were the same in relation to these matters.

### 2.2. Sample 

The number of people infected with COVID-19 was 355,956 in Metropolitan Bangkok. Moreover, there were 46,047 cases in Nakhon Ratchasima province, 37,892 cases in Ubon Ratchathani province, and 13,150 cases in Nakhon Phanom province [3], totaling 453,045 cases.

Convenience and snowball sampling techniques were used. The inclusion criteria were Thai adults aged ≥18 years who had been infected with COVID-19 and recovered within ≤6 months—not only the hospitalization patients but also non-hospitalization patients with negative test results.

They were not cognitively impaired, were able to communicate by using social media with no physical suffering, and were willing to provide rich information. The inclusion criteria of this study was written informed consent. The exclusion criteria included cognitive impairment and visual and hearing impairment. Participants (key informants) comprised 32 adults. These were screened, and 25 were eligible. In grounded theory, the ultimate criterion for the final sample size is theoretical saturation, which relates to the development of theoretical categories and relates to grounded theory methodology [17,18,19]. Moreover, a sample size of quantitative ones was calculated using G* power 3.1.9.4 [20], which was based on an acceptable power level of 0.95, effect size of 0.30, and alpha level of 0.01. For correlation, the estimate sample size was 152 participants. This study required a sample size of 180 participants for missing data. This number comprised 90 from the Metropolitan Bangkok and northeastern area, including 40 from Nakhon Ratchasima province, 30 from Ubon Ratchathani province, and 20 from Nakhon Phanom province. The setting was the participant’s home onsite, cellphone, and also via Google Forms. 

### 2.3. Protocol 

After the IRB approval, the principal investigator (PI) contacted the administrators of the hospital for permission (hospitalized participants) from patients, who were discharged from the hospital during 1 and 6 months to conduct the study. For non-hospitalized participants, snowball sampling techniques were used. Data were collected between February–May 2022. Qualitative data were collected using a semi-structure interview, open-ended dialogue, field notes, reflective notes, observation, and audio recording. A quantitative instrument was built based on ground data from a qualitative approach; then, the instrument feature was tested on a sample of the population [19]. This instrument divided physical distress into 19 items, psychological distress 22 into items, and coping into 12 items on a five-point Likert scale. The question score ranged from 1 (strongly disagree) to 5 (strongly agree). The score ranges are grouped into mild physical distress (≤44), moderate level of distress (45–70), and severe level of distress (≥71). The score ranges of psychological distress are mild level (≤51), moderate level (52–81), and severe level distress (≥82). Then, the score ranges of coping are mild level (≤28), moderate level (29–45), and high level (≥46). 

#### 2.3.1. Phase 1: Qualitative Study

The sequential exploratory strategy involved a first phase of a qualitative part that was based on a grounded theory approach [17]. Rapport and trust were established first. The data were collected through semi-structure interviews in private or at the participant’s home, which was well ventilated. The researchers reviewed the research procedure and obtained informed consent from the participants. Each interview lasted around 30–45 min, two to four times. Qualitative data was initially collected from all interviews and were conducted on site, and by video call and audio-recording. In-depth interviews were guided by the research questions. Some examples of questions were: “Could you please describe your experiences while infected with COVID-19?” “Could you please describe the physical symptoms while you were getting ill?” “How did you feel? Was there any emotional effect?” Observations were focused on their voices and the expressions of experiences. These interviews were conducted face-to-face and also using cellphones. During the interviews, family members or significant others were with the participants. The data were saturated when the information collected in the study became redundant or fresh data no longer sparked new insights [19]. Therefore, a grounded theory has systematic steps involving generating categories of information (open coding), selecting one of the categories and positioning it within a theoretical model (axial coding). Additionally, there is the explication of a story from the interconnection between these categories or selective coding [17,19]. The study was conducted between February and May 2022.

#### 2.3.2. Phase 2: Building an Instrument

A new instrument was built following the first phase because an existing instrument was not available [17,19]. The content was validated by five participants. Therefore, the instrument was taken back to five participants for judging the accuracy and credibility of their accounts [17,19]. Then, it was pilot-tested for reliability among 30 people infected with COVID-19. The Cronbach’s alpha coefficient was 0.74.

#### 2.3.3. Phase 3: Quantitative Study

This study focused on the subsequent quantitative phase. The data was collected using either Google Forms via online surveys or a conventional questionnaire onsite, with a total of 180 participants. In this phase, an entirely different sample from the first phase was used [19]. 

### 2.4. Data Analysis

A mixed methods approach combined qualitative content analysis of the interviews and statistical analysis of the survey. The interviews were recorded and transcribed to be subsequently analyzed by using the software ATLAS.ti 22 (Thaiware Communication Co., Ltd., Bangkok, Thailand) and was applied by using the Strauss and Corbin method as cited in Creswell [17,19]. A grounded theory was chosen because it is to move beyond description and to generate or discover a theory and a unified theoretical explanation [17,19,21]. Those were really physical and psychological distress and coping strategies among people infected with COVID-19. The theory was derived from the words of the key informants, which were not derived from another theoretical framework [17,19]. An iterative process of data analysis was used to develop a theoretical explanation of grounded human behavior [22]. The researchers performed all data analysis tasks with regular consultation and feedback from three external auditors to review the entire project [19]. It was a three-phase approach including (1) researchers gathering qualitative data and (2) using the analysis to develop an instrument, which was (3) subsequently administered to a sample of the population [17,19]. Then, themes were searched, mapped, and interrelated so that the meanings of the themes could be interpreted. Quantitative data were analyzed using the software SPSS version 18 (IBM Corp. Released 2019. IBM SPSS Statistic for Windows, Version 28.0 Armonk, NY, USA: IBM Crop.). Descriptive statistics, Pearson’s correlation, and the independent *t*-test were used.

#### Rigor and Trustworthiness

Four criteria were considered—credibility, dependability, confirmability, and transferability—and were accumulatively contributed to trustworthiness [17,19,21,22]. Rapport and trust were established and prolonged periods of time were spent. The interviewers avoided using ideas to lead participants to express their physical symptoms, psychological distress, and coping strategies. The reflexive data, for which the research team had checked methods, were carefully collected to establish confirmability. Dependability was enhanced through debriefing data collection and analysis including external consultants, sharing emerging ideas, codes, and interpretation. It was confirmed that the findings and conclusions were supported by the data collected. 

### 2.5. Ethical Aspects

The study was conducted according to the guidelines of the Declaration of Helsinki and approved by the Institutional Review Board of Kuakarun Faculty of Nursing, 

Navamindhradhiraj University (approval no KFN IRB 2022-2, and date of approval 17 February 2022).

Informed consent was obtained from all subjects involved in the study. They could withdraw from the study at any time without negative consequences. To protect confidentiality, pseudonyms were used by the participants. All audio-recording and transcript data were stored securely. Participants who agreed to participate in the study provided written informed consent.

## 3. Results

### 3.1. Characterization of the Study Participants

The participants’ demographic characteristics in the mixed methods, the total qualitative method, consisted of 25 participants with COVID-19 with 15 of them having severe symptoms while the other 10 had non-severe symptoms. They were 10 males and 15 females, with an average age of 52 years (ranging from 21 to 85 years old). Eight of them were admitted to the ICU, 10 to the COVID ward, and 7 were not hospitalized. The participants of the quantitative group were 180, comprising 45 severe cases (25%) and 135 non-severe cases (75%). They were 56 males (31%) and 124 females (69%)) 87 vulnerable people (aged ≥ 60, co-morbidity, and pregnant women) (48%), and 93 non-vulnerable (52%). Additionally, 32 were not vaccinated (18%) and 148 vaccinated (82%) (Table 1).

### 3.2. Mixed Method Study

In this mixed method study, the qualitative and quantitative data were integrated in each major theme. From the analysis of data, six major themes emerged with ten sub-themes (Figure 1).

In addition, a grounded theory framework was built (Figure 2).

#### 3.2.1. Theme 1: Severity of COVID-19 Symptoms

Fifteen participants with severe symptoms all expressed that they were admitted long-term to the ICU or COVID-19 ward for between 28–40 days. According to symptoms, there was a progression of infection. They stated that they had dyspnea/difficulty of breathing or shortness of breath, tiredness, fever and chills, headache, backache, sore throat, stomach upset, diarrhea, etc. Three of them were in shock and unconsciousness. They said the following: 


*“I have high fever and chills. I took medication and warm wiped to reduce fever. It helped me a bit. The nurses helped me by giving me some oxygen but it didn’t work well and after that I blacked out. When I was conscious again, a tube was inserted in my throat and connected to the monitor. There were many tubes on me. A few days later the doctor had to make a hole on the outside of my throat to help me breathe better. I was admitted into ICU for 30 days and was transferred to the primary hospital for 10 more days.”*
(S-female, 47 y)

The non-severe cases of the high-risk group, including patients of cancer, diabetes mellitus (DM), hypertension, heart disease, chronic kidney disease (CKD), obesity, and pregnant woman, were admitted to be monitored closely. They took Favipiravir, which helped them to recover very soon, by 7–14 days.

*“I had mild symptoms, having low fever, headache, body aches but, I was in a risk group I have cancer and on chemotherapy. The doctor advised me to be admitted. When I walked to the toilet, I felt very tired. I needed to have oxygen and took Favipiravir after which I felt better. I stayed in the hospital for 4 days and 14 days in home isolation.”* (N-S 10- female, 59 y)

#### 3.2.2. Theme 2: Death Anxiety

Participants had an increase in stress, fear of death and anxiety, and sadness when their symptom conditions had worsened. They were afraid of dying from lung infection. They expressed fear of facing dying alone and felt hopelessness. Two of them were pregnant women who mentioned that they feared the death of the fetus from lung problems and shortness of breath. They had experienced the death of other patients in the ICU every day and the bad news of family death, which was a situational stressor. 


*“At that moment, I was so worried and feared death. I was not ready to die and I hadn’t done anything yet about my assets and will management.”*
(S- male, 75 y)


*“For me, I had mild symptoms, took anti-viral drugs for 2–3 days. I felt much better, but my 2-year-old daughter had a fever and her ATK test was positive. Her fever wasn’t that high but she developed a seizure. I’m so sorry for her, I felt so sad and worried about her. I didn’t want to lose her and I cried a lot, couldn’t sleep. I had no time to rest. I started to have chest pain, shortness of breath and was coughing a lot. I couldn’t stand how I felt at that time. Then I thought that I had to be stronger in order to look after my daughter. Two months have passed and I’m still scared for her.”*
(NS- male, 43 y)

#### 3.2.3. Theme 3 Uncertainty 

Participants had the uncertainty of severe symptoms and unknown prognosis. The false negative results of COVID test were found and needed to be repeated 3–5 times to get a positive result. Most of them did not trust the efficacy of the COVID vaccine. They were worried about the side effects of them. There were 4 older adults with underlying diseases and 2 pregnant women hadn’t got any vaccination because there weren’t any recommendations for the testimonial of the vaccine. Some of them had lost family members. 


*“I was not expected to be infected. There were problems with both the ATK and PCR test results. I had done the test 4 times and the results were negative until the fifth time when mine was positive. In spite of the onset of the symptoms became apparent and getting worse.”*
(NS- male, 43 y)


*“I was admitted for oxygen therapy at all time. I received the bad news during my admission that my son who had cirrhosis of the liver with COVID infection died after 5 days admission to hospital.”*



*“I was so sad, I did not expect him to die so soon. It was reality uncertainty.”*
(S- female 75 y)


*“I protected myself strictly because my work was related to COVID infection patients. I wore N 95 double masks and face shield all time including washing my hands frequently. The first COVID infection was on 24 August 2021. I had got it from my workplace and Delta virus was found. On 5 March 2022, I was infected again with Omicron virus…I could not believe it.”*
(NS- male, 30 y)

#### 3.2.4. Theme 4: Barrier to Healthcare Access

Participants expressed that the Delta variant was very dangerous and spread easily. The risk groups or vulnerable groups acknowledged the danger of it, i.e., pregnant women and aging people with underlying diseases such as DM, hypertension, cancer, chronic lung disease, chronic kidney disease (CKD), etc. At that moment, the situation area was critical and in peak conditions. The people were close to panic when they focused on the number of infected and dead people and fought for rapid antigen tests and for admission as well. Participants mentioned that the official announced that there were many sick and dead people, so the hospitals in Bangkok would not be able to handle them. 


*“I was certain that the Delta virus was very dangerous, even though I protected myself really well but I still got it.”*
(NS- male, 43 y)


*“In 2021, while I was infected, the situation was so scary. There were many infected people taken by ambulances to the hospitals but there were not enough health services, not even COVID screening.”*
(S- male, 75 y)


*“I contacted many hospitals in Bangkok but there was not a place offering a COVID test and the admission wait was 3 days. I began to feel worse, so my husband with my child took me in our car up country, more than 400 km away. Luckily, I was admitted to a hospital; if not, I would have died.”*
(S- female, 47 y)

#### 3.2.5. Theme 5: Compliance and Self-Regulation Coping 

The participants expressed their experiences of coping with the onset. They performed the COVID test, using ATK by themselves, and went to hospital for their treatment. They would follow all the rules and accepted to do so because they were afraid of dying, thinking positively that everything would be better. The severe group complied with everything the healthcare provider advised, cultivating patience to suffering in the ICU. The non-severe group would do as advised by well-informed persons, using alternative therapies such as consuming herbs to decrease cough and sore throat. They were remote treated by Favipiravir sent from the hospital. Some cases used spiritual and religious methods. Moreover, social support such as the assistance of family members and health volunteers made them feel more secure. 


*“I prayed with a little Buddha image which I brought with me to the ICU that made me mindful, distractions, lay on lateral side and on oxygen therapy all the time as the nurse advised and I felt better.”*
(S- pregnant woman, 31 y)


*“The nurses advised me to lay on my stomach and I did as they advised. I could do only 5–7 min in the daytime, but the night time I was so afraid that no one could come to help me in case the oxygen tube was missing.”*
(S- female, 75 y)


*“My daughter who looked after me, called the ambulance which took me to the hospital. She had talked to the doctor via phone every day and let me know all that.”*
(S- male, 75 y)

#### 3.2.6. Theme 6: Post-COVID-19 Effects

Almost all participants remained with post-COVID-19 symptoms such as tiredness, headache, fatigue, and decreased activity tolerance. In a few cases, the sugar level was increasing and the doctor prescribed medication to control it. There was an increase in allergies, the lungs had fibrosis, and hands and feet were numb. Memory was decreased. A pregnant woman expressed that her baby has both valve stenosis and regurgitation.


*“I was discharged 6 months ago. At the moment, I am not the same. I am easily tired. My sugar level has increased and the doctor prescribed me more medication for my DM.”*
(S- female, 47 y)


*“After my baby was born by cesarean surgery his weight was 3000 g and being easy baby. The doctor told me that he had abnormal heart valves. There were both of leakage on the right side and stenosis on the tip of lung. I believed it was caused by the COVID infection but the doctor stated that the baby might have had it as a congenital disease.”*
(S- pregnant female, 29 y)

In this mixed methods study, the qualitative and quantitative data were integrated in each major theme using a new instrument. This phase examined physical distress, psychological distress, and coping experiences among people infected with COVID-19. The findings revealed that overall physical distress was at a moderate level (mean = 48.06, SD = 19.93), and psychological distress was at a moderate level (mean = 52.01, SD = 20.94). Coping was at a moderate level (mean = 37.68, SD = 13.92). 

The relationship between physical distress, psychological distress, and coping was positive (Table 2). The comparisons of mean difference of physical distress, psychological distress, and coping between non-severe and severe groups by using the independent t-test. The matching of the participants’ ages in the non-severe group and severe group were administered (45 for each equally). There was a significant difference (Table 3).

The post-COVID-19 effects during the first and sixth months were 70% effects. There were decreased activity tolerance, fatigue, dyspnea, etc. (Table 4).

## 4. Discussion

The study found that participants were women more than men. It was controversial, as another study found the number of men patients were higher than women [9]. The number of vulnerable was higher than non-vulnerable. In accordance with the study, Amin et al. [7] suggested that the hospitalization rate increased for older and comorbid patients. 

The first major theme was the severity of COVID-19 symptoms such as fatigue, dyspnea, cough, sore throat, headache, chills, etc. The previous study reported symptoms of headache, fever and chills, exhaustion, and shortness of breath, which were prevalent among more than 50% of the participants and developed within four days in above 90% of the patients [7]. The severe group was admitted to ICU with intubation, ventilator, and high-flow nasal cannula (HFNC). Using HFNC was associated with a trend toward increased survival in ICU or hospitalization [23]. However, the recommendation led intensivists to adopt an early intubation strategy to limit the use of HFNC [24]. 

Then, non-severe groups were admitted to be monitored closely and given oxygen therapy. They took antiviral drugs, which resulted in short hospitalization stays and quick recovery. The studies found that antiviral drugs exhibited efficacy to improve clinical outcomes in patients with COVID-19; none demonstrated efficacy in reducing mortality [25].

The second major theme was death anxiety, referring to a great fear of dying. It is a phenomenon related to unknown prognosis of disease, environmental stressors in the ICU, and lack of communication with healthcare providers [26]. Previous studies found that hospitalization in an ICU and intubation were sources of death anxiety in COVID-19 patients. The results of studies show that the rate of death anxiety in patients with COVID-19 is very high, and the patient suffers from a high level of death anxiety [26]. Previous studies suggested that death anxiety was relatively high during the COVID-19 pandemic process [27]. Patients with COVID-19 experience great psychological distress during the acute phase of the disease or even long after recovery [28]. Other studies showed that patients with COVID-19 would experience a difficult time during isolation because of physical problems, loneliness, and being separated from the family [29]. Prevalence of psychological distress increased sixfold from 9.6% to 72.5% [30]. Thus, fear and thinking about the death of a patient with COVID-19 is a significant challenge that should be attended to.

The third major theme was uncertainty; although a natural and unavoidable part of life, it is also a prevailing cognitive state during health crises, as in the case of the ongoing pandemic [31]. The uncertainties associated with COVID-19 include uncertainty at the level of the disease and its management, including its prognosis and how health- and social-care systems and professionals who work within them should respond [32]. The study reported that nursing care areas would likely be missed for patients with SARS-CoV-2, and also barriers to delivering care [33]. Most of the participants expressed that the Delta variant was very dangerous and spread easily. The study found that the severity of the SARS-CoV-2 Delta variant was 11.1%. The risk of progression to severe cases increased 13.44-fold and 3.92-fold when the age was greater than 58.5 years [34]. The Delta variant also poses a greater risk of more severe outcomes, including in younger age groups [35]. Therefore, SARS-CoV-2 included a considerably more significant, more deadly, risk of hospitalization and intensive care. 

The fourth major theme was the barrier to healthcare access related to the situation of overcrowding of infection and in the hospital as reported via the government hospital on 21 June 2021. In Thailand, there was a shortage of beds, even for severe cases.

The country’s total fatalities have grown. Some have died in their home because no hospital beds were available. Moreover, others have died on the streets of Bangkok [36]. These were situational dilemmas. Similarly, reports of the situation getting worse because of insufficient beds for patients in the capital is an issue that needs urgent action to reduce new infections and requires establishing ICU rooms in field hospitals [13]. Previous studies found that problems and challenges of a nursing administration model included the structure and management of nursing manpower. The study suggested that the hospital’s nursing administration model that was developed during the COVID-19 outbreak for the in-patient department was effective and practical in managing the care of COVID-19 patients [37]. Moreover, studies in Thailand found that the four main impacts of the COVID-19 pandemic on professional nurses were (1) work-life imbalance because of increased workload, (2) fear of infection and transmission, (3) inadequate organization support including supply of PPE and quality vaccines, information support, and unfair compensation in some hospitals, and (4) ecological changes in both positive and negative directions [16].

The fifth major theme was compliance and self-regulation. The participants complied with the advice of healthcare providers regarding fear of death, the cultivation of patience, and strengthening the mind together with positive thinking that everything will be fine. They also used spiritual and religious methods, mindfulness, and transcendence meditation. Self-regulation seems to be a good indicator of adopting a healthy lifestyle and better mental health and well-being in the context of the COVID-19 pandemic [38]. A previous study found that COVID-19 distress and general mental distress were strongly related; both meaningfulness and self-control were negatively associated with general mental distress [39]. Social support from family, friends, community, and the workplace may be protective. The study found that social support moderated the relationship between perceived uncontrollability and mental health symptoms [40] and was associated with lower COVID-19 psychological impact, though not with threat perception [41]. Previous studies found that psychological distress during the COVID-19 pandemic was about having a poor self-perception of health, having diarrhea, headache, muscle pain, and having had casual contact with an infected person [42]. Moreover, the study found that resilience had a positive relationship with mental health, and social support served as a buffer against the negative impact of low resilience on mental health [43]. 

The sixth major theme was post-COVID-19 effects experienced from one to six months, the residual of physical symptoms and psychological symptoms (70%). They had two more persistent post-COVID-19 symptoms (54%). A previous study revealed that survivors had persistent symptoms, depression, and dysfunctions of specific organs, mainly the lungs, heart, kidneys, and nervous system [44]. The relationship of physical distress, psychological distress, and coping were positive relating. The severe group’s psychological distress, and coping was significantly different with the non-severe group. This study found that 72% had decreased activity tolerance, 60% fatigue, 60% anxiety/fear abnormal lungs, 50% dyspnea respectively, etc. Previous studies suggested that female patients and those with more severe initial illness were more likely to suffer from the sequelae after one year [45]. The prevalence was higher in women than men [46]. The most frequent persistent symptoms were fatigue (44.6%), smell impairment (27.7%), and dyspnea (24.09%) [46]. They had one or more long-term symptoms. The five most common symptoms were fatigue (58%), headache (44%), attention disorder (27%), hair loss (25%), and dyspnea (24%) [47]. Similarly, several studies found that the most common symptoms were fatigue, attention disorder, hair loss, dyspnea, anxiety, stress, depression, and worse sleep quality [44,45,46,48]. 

Moreover, patients experienced dysfunctions of specific organs, mainly the lungs, heart, kidneys, and nervous system [44]. Women were at a higher risk for developing long-term post-COVID symptoms including anxiety, depression, poor sleep quality than men [49].

That has further negative impacts on the quality of life. 

## 5. Limitation

The participants in this study were people infected with COVID-19 in the Metropolitan Bangkok area and three provinces in the northeastern area of Thailand. The sample was convenient, and the snowball sampling technique totaled 180 participants. Therefore, it could not be generalized to a wider scale in Thailand. The non-severe cases were more than the severe cases. In this study, the first instrument was developed from grounded data in the initial phase. Finally, the development of an instrument and factor analysis is needed.

## 6. Conclusions 

The findings supported 6 themes related to people infected with COVID-19, which included 10 sub-themes. This study adds a great emphasis on physical distress, psychological distress, and coping. Overall, physical distress, psychological distress, and coping were at a moderate level. The physical distress, psychological distress, and coping were positive relations. The severe group had psychological distress and coping higher than non-severe group. The prevalence of post-COVID-19 effects (long COVID-19) were many residual symptoms, including decreased activity tolerance, fatigue, dyspnea, anxiety and fear of reinfection, impairment of lung, heart, liver, kidney, etc., which was higher in women than men. The healthcare providers should be concerned with sufficient of healthcare services, physical and psychological support during illness, and post-COVID-19 effects. Interventions are needed for the improvement of physical and mental health of people infected with COVID-19.

## Figures and Tables

**Figure 1 ijerph-19-14657-f001:**
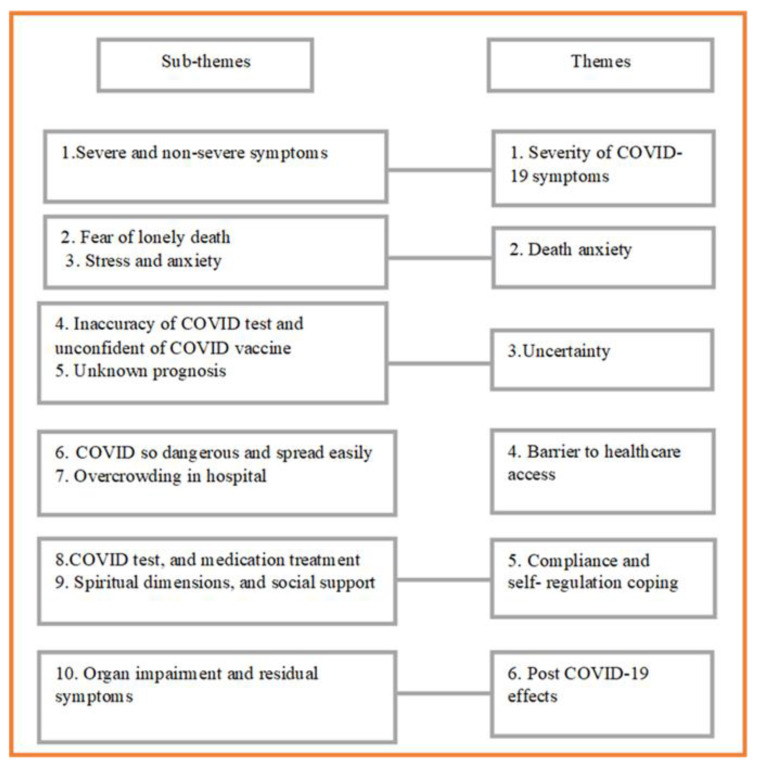
Themes and subthemes: physical, psychological distress, and coping among people infected with COVID-19.

**Figure 2 ijerph-19-14657-f002:**
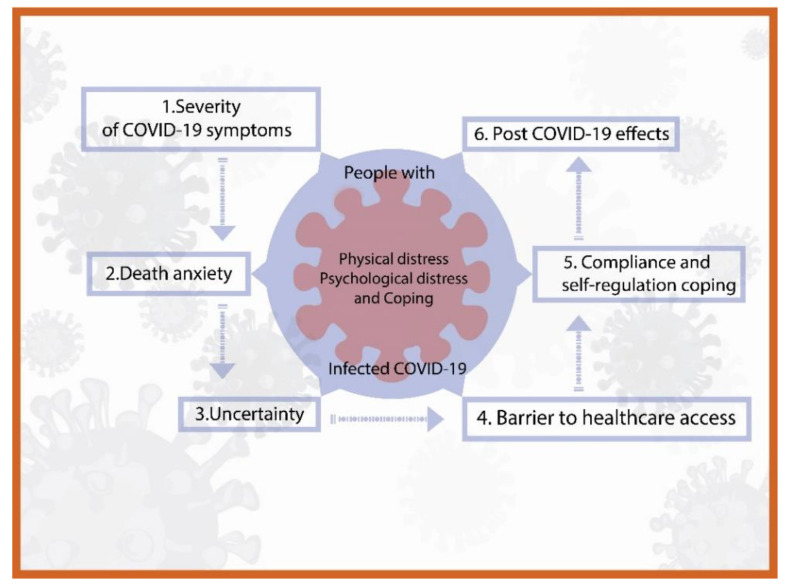
A grounded theory framework: physical, psychological distress, and coping among people infected with COVID-19.

**Table 1 ijerph-19-14657-t001:** Participants demographic characteristics.

Participants Demographic Characteristics	Frequency	Percent
Non-aged (≤60 year)	133	73.9
Aging/older adult (≥61 year)	47	26.1
Severity		
Severe	45	25
Non-severe	135	75
Gender		
Male	56	31.1
Female	124	68.9
Educational		
Elementary school	89	49.4
High school or vocational school	30	16.7
Bachelor degree, and higher degree	61	33.9
Employment Status		
Retirement	34	18.9
Employed	84	46.7
Unemployed	31	17.2
Student	20	11.1
Housewife	11	6.1
Vulnerability/Comorbidity		
No	87	48.3
Yes	93	51.7
Comorbidity	87	63.5
Pregnant	3	2.2
Older adult	47	34.3
Hospitalization		
No	95	69.3
Yes	75	54.7
Vaccination		
No	32	18
Yes	148	82

**Table 2 ijerph-19-14657-t002:** Correlation between physical distress, psychological distress, and coping among people infected with COVID-19 (n = 180).

Variables	1. Physical Distress	2. Psychological Distress	3. Coping
1. Physical distress	1	0.62 **	0.18 *
2. Psychological distress	0.62 **	-	0.33 **
3. Coping	0.18	0.33 **	1

Note: * = correlation is significant at the level 0.01, ** = correlation is significant at the level 0.001.

**Table 3 ijerph-19-14657-t003:** Comparisons of mean difference of physical distress, psychological distress, and coping between non-severe and severe groups (n = 90).

Variables	Non-Severe Group n = 45		Severe Group n = 45		t-Test	*p*-Value
	Mean	SD	Mean	SD		
Physical distress	48.06	13.98	71.02	7.16	−14.66	<0.001
Psychological distress	38.71	16.31	66.15	19.09	−7.33	<0.001
Coping	34.46	13.17	40.68	12.88	−2.26	<0.05

**Table 4 ijerph-19-14657-t004:** Post-COVID-19 effects (n = 180).

Effect of COVID-19	Frequency	Percent	95% CI (Lower–Upper)
Non-effects	54	30	23.6–36.7
Effects	126	70	63.3–76.4
1. Decreased activity tolerance	72	40	32.8–47.2
2. Fatigue	60	33.3	25.3–39.4
3. Anxiety/fear of abnormal lungs	60	33.3	25.3–39.4
4. Dyspnea	50	27.8	21.4–31.4
5. Allergy and asthma	44	24.4	18.9–31.1
6. Lung impairment	40	22.2	16.1–28.8
7. Cough/sore throat	23	12.8	7.8–18.0
8. Muscle pain/back pain/and neck pain	23	12.8	7.8–18.0
9. Drowsiness	23	12.8	8.3–19.1
10. Sleep disturbance	22	12.2	6.7–16.9
11. Loss of appetite	22	12.2	7.5–17.5
12.Headache	13	7.2	3.9–11.1
13.Hypertension (HT)	12	6.7	3.6–11.1
14. High blood-sugar level	12	6.7	3.3–10.3
15. Memory impairment	10	5.6	2.5–9.4
16. Palpitation/chest pain	8	4.4	1.7–7.5
17. Hair loss/fragile nail	6	3.3	1.1–5.6
18. Hyper platelet and stroke	3	1.7	0.0–3.3
19. Eyes impairment	2	1.1	0.0–2.8
20. Effect more than 2 symptoms	98	54	47.2–61.7
21.The others: liver, kidney impairment, gout, and neuritis	12	6.7	3.3–10.6

Note: Many of them had more than two effects.

## Data Availability

Not applicable.

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
