# Peer review of "Physical as Well as Psychological Distress and Coping with Situational Dilemmas Experienced by People Infected with COVID-19: A Mixed Method Study"

_ijerph, 2022, doi:10.3390/ijerph192214657_

Round 1

Reviewer 1 Report

This is very well written paper. The methodology is clearly described (both qualitative and quantitative methods are used) and the data analysis and the interpretation of the findings are supported by rich empirical material from this mixed methods study. The principles of grounded theory allow for the application of an inductive approach that highlights the physical, psychological distress, and coping experience with COVID 19 in the Tai context.

I have just a few suggestions for additional reflections on certain themes if they emerge in the interviews. In my mind, it would be interesting to discuss if the participants in the study expressed their attitudes to the vaccines against COVID 19 and how their experience with the disease influenced their perceptions on the vaccines and other prevention methods. Have there been any anxieties experienced by interviewees about the treatment and pharmaceutical products and procedures used in this process? Are there any ambiguous views about the severity of the disease or about the effectiveness of treatment and protection? It is also interesting to know how the participants described the difficulties and their anxieties with the treatment at home and care they received by close people. I realize that these are additional perspectives of the analysis but some of them may be added as short reflections if they are discussed by the interviewees.   

Author Response

Dear reviewer

Thank you very much for your kindness, and suggestions. Please find the file attachments.

  • Added more discussion

Sincerely,

Wichitra Kusoom

Reviewer 2 Report

Physical as Well as Psychological Distress and Coping with 2 Situational Dilemmas Experienced by People Infected with 3 COVID-19: A Mixed Method Study

 The manuscript is a mixed study that uses qualitative research and quantitative research. In this respect, the study is innovative. The following changes are suggested.

It is recommended not to use the acronym PhD meaning physical distress, because it gives rise to confusion with Ph.D. having the meaning of doctorate.

This sentence is not well understood "Similar to the globe, Thailand's population is 66.1 million [1], that had approximately 4.4 million confirmed novel Corona Virus (COVID-19) cases with 29,681 deaths s [2]. ·  On the other hand, a verb is in the present tense and another in the past tense. Both verbs should both be in the same tense.

Authors should remember that the reader does not have to be familiar with the subject, so it should be explained briefly in the grounded theory introduction and provide bibliographic references.

In the material and methods, it would be interesting to include two sections, one for qualitative and one for quantitative analysis.

It is indicated that a questionnaire has been developed, and Cronbach's alpha is provided. The authors should provide more detailed information on preparing the questionnaire.

The authors used a questionnaire they designed with physical distress 19 items, psychological distress 22 items, and coping 12 items. Please include as additional material the English translation of the questionnaire.

In line 122, the authors say that they used the Strauss and Corbin method. The authors should include in this place of the text a reference to the method.

The text of patient interview transcripts is in italics, this is correct, but it should also be in quotation marks.

The text indicates that Cronbach's alpha had a value of .74. It should be 0.74. Similarly, similar changes should be made throughout the article.

Table 1 contains numbers where a 0 should be included before the point. For example, .18 should be 0.18.

There are several numbers with asterisks in Table 1 but in no case is it explained what it means.

Table 1 should indicate in the title that what is presented is the Pearson correlation coefficient.

Table 1 is not well designed. It is not understood what the headers of the columns refer to (1,2,3), it can be guessed that they are the names of the columns, but this is misleading. In any case, it must be modified.

It is not understood why the copying row does not show any coefficient. Table 1  needs to be improved.

In table 2, which stands for ES = Effect Size?

In table 3, the title should be somewhat more informative, for example, side effects of covid19 declared by patients or another phrase that seems better to the authors.

They should include the confidence interval of the percentages in Table 3. This can be quickly done with the freely available openepi program https://www.openepi.com/Proportion/Proportion.htm

Statistical analysis is poor. Only Cronbach's alpha is present. The data have been little exploited, and more attention should be paid to the questionnaire's psychometric properties so that other researchers can validate and use it in the future. It would be interesting to present a factor analysis of the questionnaire and the results in a table. It is suggested to include more parameters, such as the test of the Parallel-form reliability or the split-half reliability.  . These calculations can be done quickly and easily with SPSS.

Table 3  presents several side effects that patients have. Are these effects comparable to those in the literature? It should be framed by giving comparative bibliographic references with studies conducted in Thailand or other countries.

Author Response

Reviewer #2

1 Changed PhD to PhyD, PsD to PsyD

  1. Changed sentence to the same tense

  1. explained briefly in the grounded theory introduction and provide bibliographic

  1. in the material and methods, added qualitative and quantitative

  1. giving more details on preparing the

  1. including as additional material the English translation of the

  • used the Strauss and Corbin method, added reference

  1. giving quotation marks “……… ”

  1. The text indicates that Cronbach's alpha had a value of .74. changed 0.74. and throughout the

  1. added new Table 1

  1. changed table 1 to table 2 , 2 to 3, 3 to 4

  1. changed .18 to 18

  1. changed table 1 (to table 2)

  1. ES = Effect Size? (table 4 was deleted),

  1. added confidence interval of the percentages in Table

  1. Statistical analysis is only Cronbach's alpha is present. But internal validity was made by taking back to 5 participants for judging the accuracy and credibility of account (described 7).

  1. The factor analysis is needed (described in limitation 22).

  1. Table 3 (=table 4) presents several side effects that patients have, explained in discussion 21-22).

  1. COVID-19 effects comparable to those in the literature? It should be framed by giving comparative bibliographic references with studies conducted in Thailand and other

Reviewer 3 Report

Thank you for the opportunity to read this very interesting paper. The paper has many strengths that should be of interest to the journal audience. Thus, the following suggestions are around enhancing the presentation for publication and clarifying aspects of the data and reporting.

I will go by line number for the most part. If not, I will try to be as specific as possible in noting the area I am speaking about.

Abstract

Purpose. It is necessary to include the country.

Methods. General population?

Keywords. Physical and psychological distress are not MeSH Terms. I recommend including mental health as MeSH term.

 1. Introduction

[Line 33] novel Corona Virus (COVID-19). Please, review the meaning of COVID-19.

Introduction section is incomplete, I think. Authors must speak more about gender and professionals healthcare, the adverse conditions affecting mental health, which are the consequences of high levels of psychological distress, etc. Is there in these environments a high presence of symptomatology related to work stress (physical and emotional fatigue, overload, tension, and anxiety) that may pose a risk of impaired mental health? Why? Please, provide a definition or a theory about the main concepts.

2. Materials and Methods

[Lines 55-56] When you say “A qualitative procedure was 55 based on a grounded theory…”, Could you say what is it?

How was the sample chosen? Which is the total population? Authors must specify it.

Authors must specify it.

Do the authors have a study protocol? The study protocol should be described in detail.

I am not sure where did the authors get the sample? Hospital? Please, describe thoroughly how the data collection process was carried out. Authors must specify it.

Why did not you use a validated scale to evaluate Psychological Distress?

In relation to the items that were created "ad hoc", it is also necessary to better describe how these items were agreed (literature review, expert consensus, etc.).

3. Results

Demographic data on respondents should be given

At last, but not least, I recommend you to make available your data in an open repository. I think it will make this scientific process more transparent, and it allows other researchers to replicate your results.

4. Discussion

The prevalence and incidence of COVID-19 in the area of study during the period of study should be discussed.

Authors need to argue their results better and they should be compared with other results.

Moreover, some points were not discussed, i.e., the participants were assessed from February and May, 2022, since a fatigue scenario could exist due to Covid-19 social restrictions; how could this factor impact these participants?

Limitations related with the type of methodology used. Limitations regarding representativeness of respondents should be better addressed Authors must specify it. The fact of having a convenience sample should be included in the limitations of the study.

Wang Y, Kala MP, Jafar TH. Factors associated with psychological distress during the coronavirus disease 2019 (COVID-19) pandemic on the predominantly general population: A systematic review and meta-analysis. PLoS One. 2020 Dec 28;15(12):e0244630. doi: 10.1371/journal.pone.0244630. PMID: 33370404; PMCID: PMC7769562.

Vlake JH, Wesselius S, van Genderen ME, van Bommel J, Boxma-de Klerk B, Wils EJ. Psychological distress and health-related quality of life in patients after hospitalization during the COVID-19 pandemic: A single-center, observational study. PLoS One. 2021 Aug 11;16(8):e0255774. doi: 10.1371/journal.pone.0255774. PMID: 34379644; PMCID: PMC8357130.

Domínguez-Salas S, Gómez-Salgado J, Andrés-Villas M, Díaz-Milanés D, Romero-Martín M, Ruiz-Frutos C. Psycho-emotional approach to the psychological distress related to the COVID-19 pandemic in Spain: a cross-sectional observational study. Healthcare. 2020; 8(3):190.

Zhang J, Lu H, Zeng H, Zhang S, Du Q, Jiang T, Du B. The differential psychological distress of populations affected by the COVID-19 pandemic. Brain Behav Immun. 2020 Jul;87:49-50. doi: 10.1016/j.bbi.2020.04.031. Epub 2020 Apr 15. PMID: 32304883; PMCID: PMC7156946.

Gómez-Salgado J, Domínguez-Salas S, Romero-Martín M, Ortega-Moreno M, García-Iglesias JJ, Ruiz-Frutos C. Sense of coherence and psychological distress among healthcare workers during the COVID-19 pandemic in Spain. Sustainability. 2020; 12(17): 6855.

Rahman MA, Rahman S, Wazib A, Arafat SMY, Chowdhury ZZ, Uddin BMM, Rahman MM, Bahar Moni AS, Alif SM, Sultana F, Salehin M, Islam SMS, Cross W, Bahar T. COVID-19 Related Psychological Distress, Fear and Coping: Identification of High-Risk Groups in Bangladesh. Front Psychiatry. 2021 Aug 13;12:718654. doi: 10.3389/fpsyt.2021.718654. PMID: 34484005; PMCID: PMC8414638.

I wish you all the best.

Author Response

Dear reviewer

Thank you very much for your kind, and suggestions. Our work was finished on revision. Please find the file attachment too.

  1. Review the meaning of COVID-19.
  2. Completed introduction section
  3. Described more about gender and professionals healthcare, the adverse conditions affecting mental health, which are the consequences of high levels of psychological distress,
  4. Added environments a high presence of symptomatology related to work stress (physical and emotional fatigue, overload, tension, and anxiety)
  5. definition or a theory about the main

Materials and Methods

  1. “A qualitative procedure was 55 based on a grounded theory…revised
  2. Added a study protocol (described in detail)
  3. Described how the data collection process
  4. An instrument validated by 5 participants for judging the accuracy and credibility of account, which based on Creswell [ see citation 17,19]. It was built following the first phase because the existing instrument was not available [17,19]

  1. It was pilot-tested for reliability with 30 people with infected COVID-19. The Cronbach’s

alpha 0.74

  1. In relation to the items that were created "ad hoc", it is also necessary to better describe how these items were agreed (literature review, expert consensus, etc.)….(see 9-10.)

Results

At last, but not least, I recommend you to make available your data in an open repository. I think it will make this scientific process more transparent, and it allows other researchers to replicate your results.

  1. Demographic data on respondents was given (table 1).

Discussion

  1. Discussed the prevalence and incidence of COVID-19 in the area of study during the period of study
  2. Discussed argue, and associated with other
  3. More discussed due to Covid-19 social restrictions….p. 23
  4. Added limitation of this study
  5. Checked citation /References
  • ·        Authors’ Contributions; Revised

Kind regards, Wichitra Kusoom

Round 2

Reviewer 3 Report

All my recommendations were implemented.